DATA RELEASE

# Improvements to the Gulf pipefish *Syngnathus scovelli* genome

Balan Ramesh[1,*], Clay M. Small[2,3], Hope Healey[2], Bernadette Johnson[1], Elyse Barker[1], Mark Currey[2], Susan Bassham[2], Megean Myers[1], William A. Cresko[2,3,†] and Adam Gregory Jones[1,*,†]

1 Department of Biological Sciences, University of Idaho, Moscow, ID 83844, USA
2 Institute of Ecology and Evolution, University of Oregon, Eugene, OR 97403, USA
3 Presidential Initiative in Data Science, University of Oregon, Eugene, OR 97403, USA

## ABSTRACT

The Gulf pipefish *Syngnathus scovelli* has emerged as an important species for studying sexual selection, development, and physiology. Comparative evolutionary genomics research involving fishes from Syngnathidae depends on having a high-quality genome assembly and annotation. However, the first *S. scovelli* genome assembled using short-read sequences and a smaller RNA-sequence dataset has limited contiguity and a relatively poor annotation. Here, using PacBio long-read high-fidelity sequences and a proximity ligation library, we generate an improved assembly to obtain 22 chromosome-level scaffolds. Compared to the first assembly, the gaps in the improved assembly are smaller, the N75 is larger, and our genome is 95% BUSCO complete. Using a large body of RNA-Seq reads from different tissue types and NCBI's Eukaryotic Annotation Pipeline, we discovered 28,162 genes, of which 8,061 are non-coding genes. Our new genome assembly and annotation are tagged as a RefSeq genome by NCBI and provide enhanced resources for research work involving *S. scovelli*.

**Subjects** Genetics and Genomics, Evolutionary Biology, Marine Biology

**Submitted:** 04 November 2022

\* Corresponding authors. E-mail: ramesh@uidaho.edu; adamjones@uidaho.edu

† Contributed equally.

Preprint submitted at https: //doi.org/10.1101/2023.01.23.525209

## DATA DESCRIPTION

This article presents a resource (genome assembly) that marks a technological improvement compared to the one previously published in the article, "The genome of the Gulf pipefish enables understanding of evolutionary innovations" [1].

A *de novo* genome assembly is evaluated based on three primary criteria: accuracy or correctness, completeness, and contiguity [2, 3]. Typically, the correctness of a genome is one of the most challenging features to measure. However, with modern, long-read sequencing technologies, the orientation of the contigs and the gene order of an assembly are highly accurate [4–6]. On the other hand, completeness and contiguity are easier to measure [6–8] yet more challenging to achieve, especially in non-model organisms. The Gulf pipefish (*Syngnathus scovelli*, NCBI:txid161590, fishbase ID: 3306) genome is an essential resource for the study of comparative genomics, evolutionary developmental biology, and other related topics [1, 9–15]. Given the technological constraints when it was initially sequenced, the first version of the *S. scovelli* genome is highly accurate and mostly complete, but it leaves considerable room for improvement with respect to contiguity [1]. Here, with the use of third-generation sequencing technology, including PacBio High Fidelity (Hi-Fi) long reads from circular consensus sequences (CCS) and Hi-C proximity ligation from Phase Genomics, we produced a nearly complete chromosome-scale genome

**Table 1.** Contiguity metrics from QUAST for various *Syngnathus* species.

| Metrics | S. acus | S. rostellatus | S. typhle | S. floridae | S. scovelli _v1 | S. scovelli _v2 |
|---|---|---|---|---|---|---|
| Number of contigs | 87 | 8,935 | 526 | 6,895 | 886 | 526 |
| Largest contig | 28,444,102 | 856,273 | 9,665,359 | 61,807,209 | 23,505,159 | 30,098,933 |
| Total length | 324,331,233 | 280,208,023 | 313,958,489 | 303,298,972 | 305,995,683 | 431,750,762 |
| Reference length | 324,331,233 | 324,331,233 | 324,331,233 | 324,331,233 | 324,331,233 | 324,331,233 |
| GC (%) | 43.46 | 43.08 | 43.29 | 43.63 | 42.95 | 45.00 |
| Reference GC (%) | 43.46 | 43.46 | 43.46 | 43.46 | 43.46 | 43.46 |
| N50 | 14,974,571 | 88,962 | 3,046,963 | 7,845,045 | 12,400,093 | 17,337,441 |
| NG50 | 14,974,571 | 70,018 | 3,012,268 | 7,783,711 | 11,493,655 | 20,118,474 |
| N75 | 11,896,884 | 34,357 | 1,098,273 | 21,150 | 8,458,319 | 13,347,818 |
| NG75 | 11,896,884 | 15,229 | 998,421 | 17,023 | 7,908,134 | 15,901,424 |
| L50 | 8 | 812 | 30 | 5 | 10 | 10 |
| LG50 | 8 | 1,092 | 32 | 6 | 11 | 7 |
| L75 | 14 | 2,068 | 72 | 1,160 | 17 | 17 |
| LG75 | 14 | 3,492 | 79 | 2,003 | 19 | 12 |

For NGx and LGx calculations, *S. acus* was used as the reference species. All the *Sygnathus* genomes (except *S. scovelli*) were last accessed from NCBI on 2022-July-26.

assembly that not only improves completeness and accuracy but is also the most contiguous genome yet produced for the genus *Syngnathus* (Table 1).

## Context

Evolutionary novelties are widespread across the tree of life. However, the origin of *de novo* genes and their associated regulatory networks, as well as their effects on the phenotype, remain mysterious in most species. Syngnathidae is a family of teleost fishes that includes pipefishes, seahorses, and seadragons [1, 12–16]. Syngnathid fishes are known for their evolutionary novelty with respect to morphology and physiology. For instance, species in this family have variously evolved elaborate leafy appendages, male brooding structures, prehensile tails, elongated facial bones, and numerous other unusual traits [1, 12–14]. With a variety of mating systems and sex roles [12–16], the syngnathid fishes also provide an excellent study system to investigate the generality of theories on sexual selection and reproductive biology [15, 16]. Advances in comparative genomics and the evolutionary developmental biology of novel traits in syngnathids require the development of additional genomic tools. Among these are well-assembled and annotated genomes [1]. Here, we took a step in this direction by producing an improved reference genome for the Gulf pipefish.

## METHODS
### DNA and RNA extraction

We collected *S. scovelli* from the Gulf of Mexico in Florida, USA (Tampa Bay), and flash froze them in liquid nitrogen. We pulverized approximately 50 mg of whole-body tissue (posterior to the urogenital opening) from a single male on liquid nitrogen, which we submitted to the University of Oregon Genomics and Cell Characterization Core Facility (UOGC3F) for high-molecular-weight DNA isolation using the PacBio Nanobind tissue kit. We submitted similar (but unpulverized) frozen tissue from the same individual fish to Phase Genomics to generate a Hi-C library using Proximo Animal (v4) technology.

In addition, we used organic extraction with TRIzol Reagent, followed by column-based binding and purification using the Qiagen RNeasy MinElute Cleanup Kit, to extract mRNA from the Brain, Eye, Gills, Muscle/Skin, Testis, Ovary, Broodpouch, and Flap tissues.

### Sequencing and assembly

After the size selection of genomic DNA using the Blue Pippin (11 kb cutoff), the UOGC3F constructed a sequencing library using the SMRTbell Express Template Prep Kit 2.0. One SMRT cell was sequenced by the UOGC3F using PacBio Sequel II technology, yielding 33.39 Gb in 2.05M CCS reads (out of 6.298M Hi-Fi reads in total). We sequenced 70.4 Gb of paired-end 150 nucleotide reads (234.6 million in total) from the Hi-C library using an Illumina NovaSeq 6000 at the UOGC3F. The RNA sequencing libraries were prepared using the KAPA mRNA HyperPrep Kit. We sequenced 159 bp paired-end reads using Illumina Novaseq 6000 for each tissue from the RNA sequencing libraries for annotation.

Using the Hi-Fi sequences, we estimated the genome size using genomescope2 (v2.0, RRID:SCR_017014) [17] and meryl (v2.2) [18] with a default k-mer size of 21 (Figure 1). The paired-end Hi-C reads were trimmed using trimmomatic (v0.39, RRID:SCR_011848) [19] with the parameter HEADCROP:1 to remove the first base, which was of low quality. Together with the Hi-Fi sequences, we assembled the first-pass genome assembly in Hi-C integrated mode using hifiasm (v0.16.1, RRID:SCR_021069) [18] with default parameters. The First-Pass assembly refers to the first draft consensus assembly from the Hi-Fi and Hi-C data. We extracted the consensus genome from hifiasm in fasta format and assembled the contigs into scaffolds using juicer (v1.6, RRID:SCR_017226) [20]. We used the 3D-DNA (version date: Dec 7, 2016) [21] pipeline to merely order the scaffolds. The Hi-C contact map of the ordered scaffolds was visualized using juicebox (v1.9.8, RRID:SCR_021172) with no breaking of the original contigs.

### Assessment of completeness and contiguity

To compare the completeness and contiguity of the latest version of the *S. scovelli* genome against the other *Syngnathus* genomes (Figure 2), we downloaded the genome assemblies of *S. acus* (GCA_024217435.2), *S. rostellatus* (GCA_901007895.1) [22], *S. typhle* (GCA_901007915.1) [22], and *S. floridae* (GCA_010014945.1) from NCBI. We used Benchmarking Universal Single-Copy Orthologs (BUSCO v5.2.2, RRID:SCR_015008) [23] in genome mode with the actinopterygii_odb10 database (as of 2021-02-19) to evaluate the completeness of the genome. Also, we used a k-mer-based assessment using Merqury (v2020-01-29, RRID:SCR_004231. [24]) to estimate the completeness and the base error rate. We then used the Quality Assessment Tool (QUAST v5.0.2, RRID:SCR_001228) [25] to estimate Nx and Lx statistics for our assembly.

### Annotation using the NCBI Eukaryotic annotation pipeline

The NCBI Eukaryotic Genome Annotation Pipeline (v10.0) is an automated software pipeline identifying coding and non-coding genes, transcripts, and proteins on complete and incomplete genome submissions to NCBI. The core components of this pipeline are the RNA alignment program (STAR and Splign) and Gnomon, a gene prediction program. In this pipeline, the RNA-Seq reads from the various (Brain, Eye, Gills, Muscle/Skin, Testis, Ovary, Broodpouch, and Flap) tissues of multiple samples, including the *S. scovelli* individual used for Hi-Fi and Hi-C sequence data (SRR20438584–SRR20438604), were aligned to the genome. Gnomon combines the information from alignments of the transcripts and the *ab initio* models from a Hidden Markov Model-based algorithm to create a RefSeq annotation. This RefSeq annotation produces a non-redundant set of a predicted transcriptome and a proteome that can be used for various analyses. The Eukaryotic annotation pipeline is not publicly available; thus, we requested the staff at NCBI to annotate the *S. scovelli* genome.



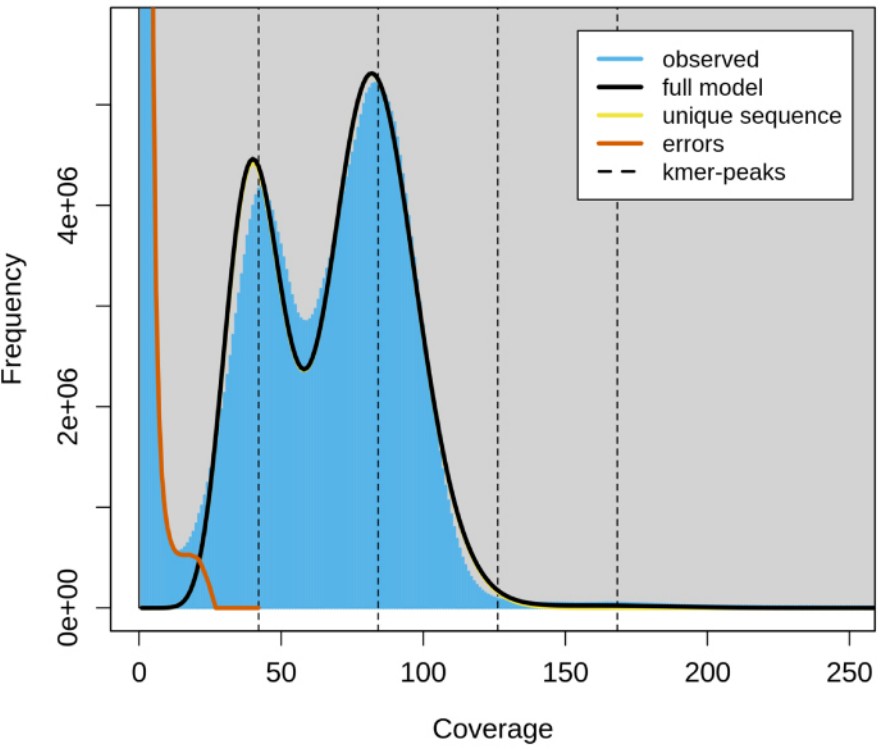

**Figure 1.** Estimated genome size of *Syngnathus scovelli* based on k-mer analysis using Meryl and Genomescope.

## DATA VALIDATION AND QUALITY CONTROL

### Assembly statistics

With approximately 2 million Hi-Fi reads and 234.6 million Hi-C reads, we generated the first pass consensus assembly with 585 contigs. The N50 and L50 for this assembly were 15.5 Mb and 11, respectively. We scaffolded this assembly to correct misassembles and produced a final assembly containing 526 contigs with N50 and L50 values of 17.3 Mb and 10, respectively (Table 1). This improved version of the *S. scovelli* genome has around three times fewer contigs compared to the original *S. scovelli* genome. The NG50 and NG75 are ~1.75× and ~2× larger, respectively, than the previous assembly, implying less fragmentation. Our new assembly reduces the number of gaps per 100 kilobase pairs (kb) from 6,837.20 Ns per 100 kb to a mere 0.27 Ns per 100 kbp, owing to the increased contiguity. This new *S. scovelli* genome is on par with the current best genome in the *Syngnathus* genus, that of *S. acus*, which is a complete chromosome-scale assembly. The first 22 scaffolds of the *S. scovelli* genome are of chromosome-scale in line with the genetic map [1] and the karyotype data [27] with a total length of around 380 Mb (Figure 3), comparable to the estimated genome size of 380 Mb (see GigaDB [28]; Table 2 and Figure 3). In addition, 88.94% of the total assembly length is captured in the 22 chromosome-scale scaffolds.

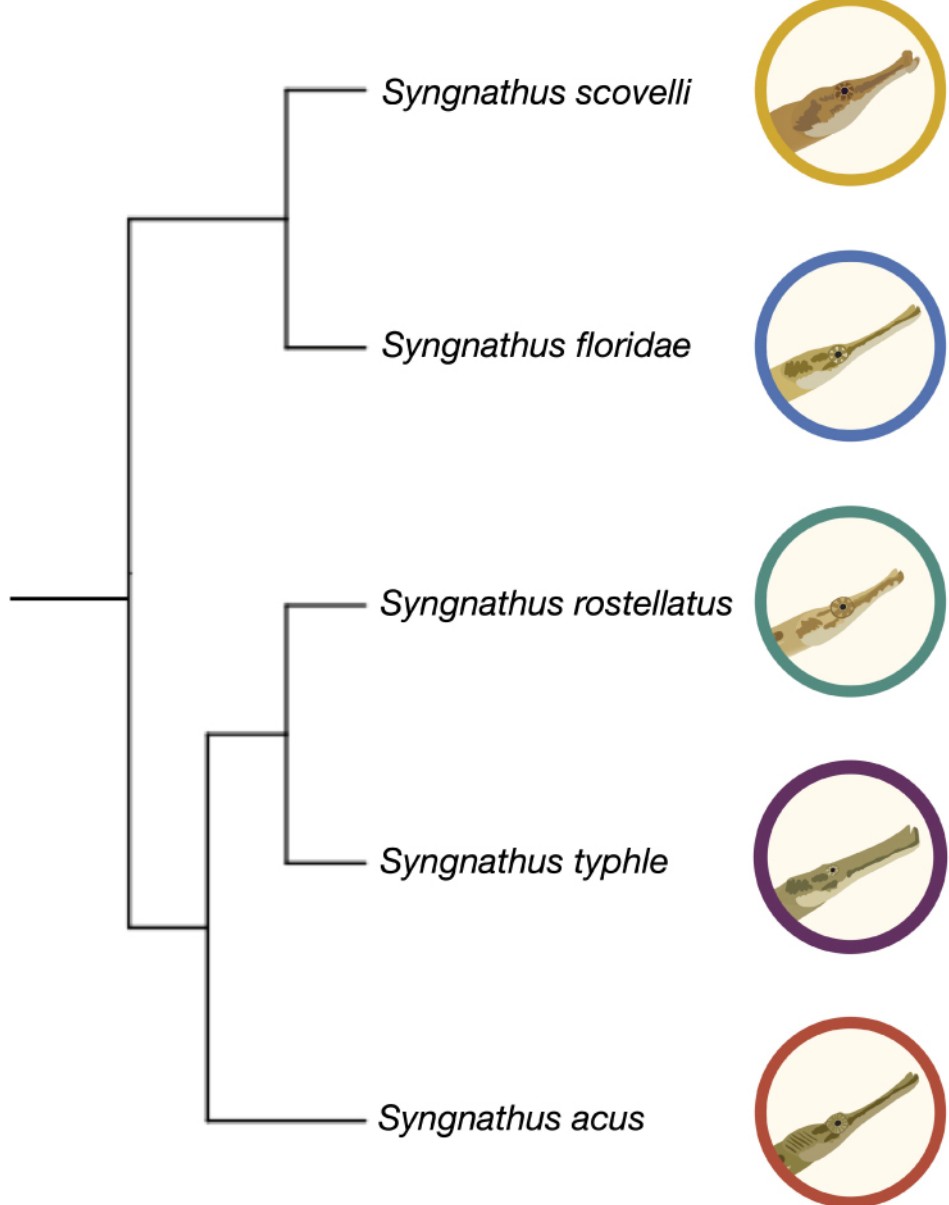

**Figure 2.** Cladogram of the five *Syngnathus* species in this study. This phylogeny is based on the Ultra Conserved Elements among all syngnathids [26].

For 15 of the chromosome-scale scaffolds, a single contig makes up the total length; the remaining seven are generally composed of a small number of contigs (Figure 3).

## BUSCO and Merqury results

BUSCO results suggest a high degree of completeness as it found 95% of the orthologs in the Actinopterygii dataset (94.7% [*S*: 93.9%, *D*: 0.8%], *F*: 1.5%, *M*: 3.8%, *n*: 3,640) when run in genome mode (Figure 4) and the Merqury evaluation suggests that the genome is ~86%

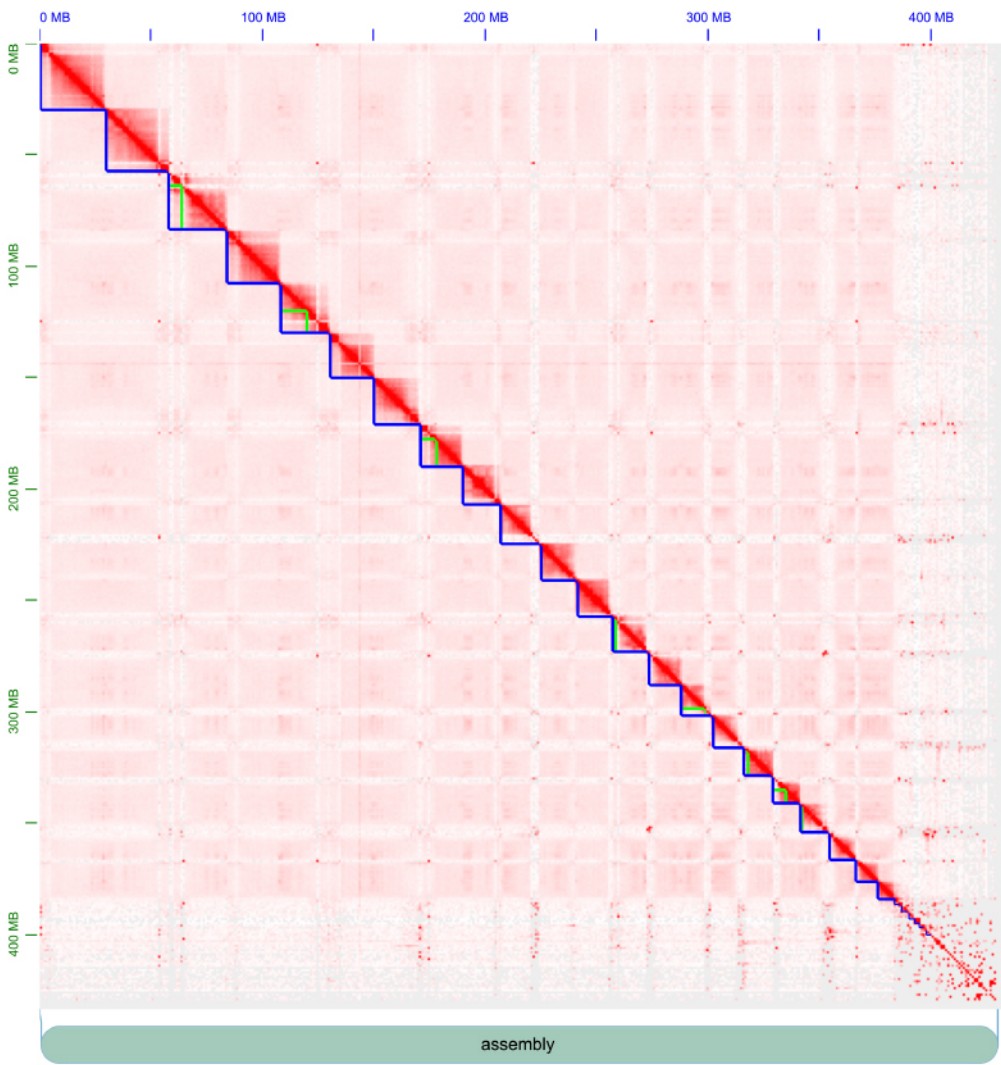

**Figure 3.** Visualization of contact maps from Hi-C reads for *Syngnathus scovelli* (v2). The first 22 primary assembly features (blue lines) sum to about 380 Mb in size, which is the estimated genome size for the species. The green lines reflect the individual contigs from the hifiasm assembly that were organized into chromosome-level scaffolds based on Hi-C contact data.

**Table 2.** Contiguity metrics from QUAST for the first pass and the scaffolded assembly of *S. scovelli* _v2.

| Metrics | Haplotype1 | Haplotype2 | Primary consensus assembly | Scaffolded assembly |
|---|---|---|---|---|
| Number of contigs | 901 | 544 | 585 | 526 |
| Largest contig | 21,671,036 | 23,661,123 | 30,098,933 | 30,098,933 |
| Total length | 427,545,154 | 428,155,884 | 431,749,582 | 431,750,762 |
| GC (%) | 44.99 | 44.78 | 45.00 | 45.00 |
| N50 | 10,825,652 | 10,535,849 | 15,551,623 | 17,337,441 |
| N75 | 4,999,310 | 4,477,557 | 11,049,644 | 13,347,818 |
| L50 | 15 | 15 | 11 | 10 |
| L75 | 29 | 30 | 19 | 17 |
| Number of N's per 100 kbp | 0.00 | 0.00 | 0.00 | 0.27 |

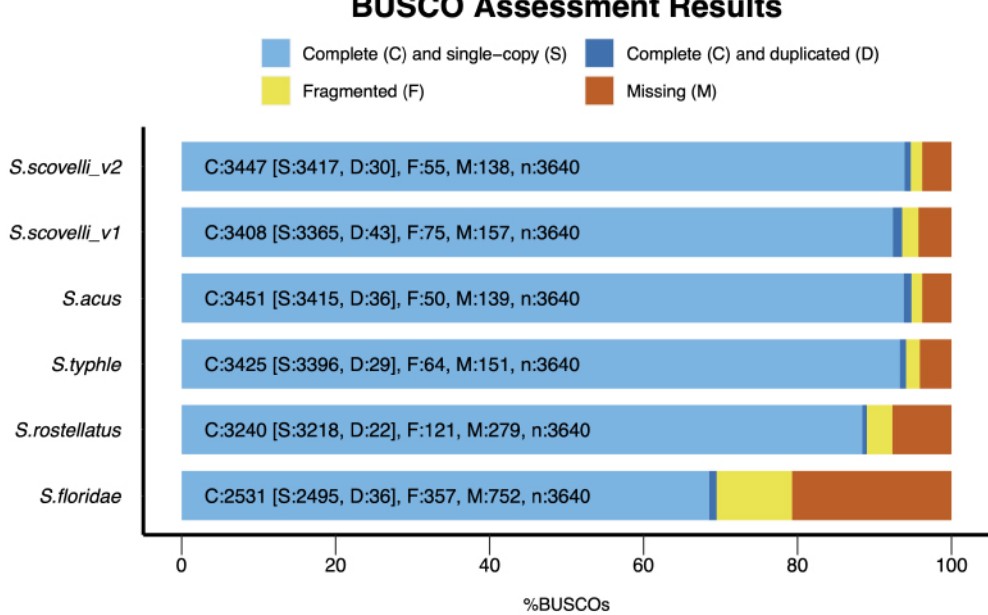

**Figure 4.** Comparison of BUSCO completeness among all the five *Syngnathus* species.

**Table 3.** k-mer based assembly evaluation for completeness using Merqury.

| Assembly | k-mer set used | solid k-mers in the assembly | Total solid k-mers in the read set | Completeness (%) |
|---|---|---|---|---|
| *S. scovelli* _v2 | all | 272,969,166 | 318,487,563 | 85.708 |

**Table 4.** k-mer based quality evaluation using Merqury.

| Assembly | k-mers uniquely found only in the assembly | k-mers found in both the assembly and the read set | QV | Error rate |
|---|---|---|---|---|
| *S. scovelli*_v2 | 6,614 | 431,737,882 | 61.3697 | $7.29504 \times 10^{-7}$ |

complete with a quality value (QV) of 61.37 and an error rate of $7.3 \times 10^{-5}$ % (see GigaDB [28] for more details; Tables 3 and 4).

Consistent with the BUSCO contiguity metrics, the genome is on par with *S. acus* for completeness, which is also around 95% complete. Missing genes make up the majority of the remaining 5% of genes. We identified genes likely to be truly missing from the *S. scovelli* genome and more broadly from members of Syngnathidae (including the seahorses, genus *Hippocampus* along with *Syngnathus*) by confirming their absence across the BUSCO results from the present assembly, four additional members of the genus *Syngnathus*, and six additional *Hippocampus* publicly available assemblies (see GigaDB [28] for additional details). Of the missing BUSCO genes, 83 are shared among all the species of *Syngnathus*, and 38 are missing from both genera (see GigaDB [28] for additional details). Future work could profitably explore these missing genes, as some may be related to the interesting novel traits in syngnathid fishes.

**Table 5.** Gene and Feature Statistics from NCBI Eukaryotic Pipeline.

| Feature | *S. scovelli_v2* |
|---|---:|
| Genes and pseudogenes | 29,062 |
| protein-coding | 20,101 |
| non-coding | 8,061 |
| Transcribed pseudogenes | 0 |
| Non-transcribed pseudogenes | 887 |
| genes with variants | 10,398 |
| Immunoglobulin/T-cell receptor gene segments | 9 |
| other | 4 |
| mRNAs | 47,846 |
| fully-supported | 47,491 |
| with >5% *ab initio* | 89 |
| partial | 39 |
| with filled gap(s) | 0 |
| known RefSeq | 0 |
| model RefSeq | 47,846 |
| non-coding RNAs | 12,092 |
| fully-supported | 7,318 |
| with >5% *ab initio* | 0 |
| partial | 5 |
| with filled gap(s) | 0 |
| known RefSeq | 0 |
| model RefSeq | 10,741 |
| pseudo transcripts | 0 |
| fully-supported | 0 |
| with >5% *ab initio* | 0 |
| partial | 0 |
| with filled gap(s) | 0 |
| known RefSeq | 0 |
| model RefSeq | 0 |
| CDSs | 47,855 |
| fully-supported | 47,491 |
| with >5% *ab initio* | 115 |
| partial | 39 |
| with major correction(s) | 144 |
| known RefSeq | 0 |
| model RefSeq | 47,846 |

## Annotation results

After masking about 43% of the genome, the annotations resulted in the prediction of about 28,162 genes, of which 8,061 are non-coding genes (see GigaDB [28]; Tables 5 and 6). The 28,162 genes produce about 59,938 transcripts, of which 47,846 are mRNA, and the rest is made up of other types of RNAs such as tRNA, lncRNA, and others. Out of the 20,101 coding genes, 18,616 had a protein with an alignment covering 50% or more of the query against the UniProtKB curated protein set, and 9,152 had an alignment covering 95% or more of the query.

## REUSE POTENTIAL

The new version of the *S. scovelli* genome opens doors to more accurate results by enhancing the comparative genome data analysis and facilitating the creation of robust tools for molecular genetic studies. We generated the original version of the genome to focus on the genetic mechanisms underlying the unique body plan among pipefishes and seahorses. This genome version takes us one step closer to uncovering these evolutionary

**Table 6.** Detailed Feature Lengths from NCBI Eukaryotic Pipeline.

| Feature | Count | Mean length (bp) | Median length (bp) | Min length (bp) | Max length (bp) |
|---|---|---|---|---|---|
| Genes | 28,166 | 11,149 | 4,361 | 56 | 677,970 |
| All transcripts | 59,938 | 3,654 | 2,773 | 56 | 106,526 |
| mRNA | 47,846 | 3,907 | 3,042 | 204 | 98,797 |
| misc_RNA | 2,018 | 3,844 | 2,824 | 138 | 22,974 |
| tRNA | 1,351 | 74 | 73 | 71 | 87 |
| lncRNA | 5,304 | 3,880 | 1,632 | 112 | 106,526 |
| snoRNA | 117 | 123 | 126 | 62 | 319 |
| snRNA | 378 | 142 | 141 | 56 | 196 |
| rRNA | 2,920 | 1,228 | 154 | 118 | 4,380 |
| Single-exon | 514 | 2,381 | 1,944 | 358 | 21,617 |
| coding | 514 | 2,381 | 1,944 | 358 | 21,617 |
| CDSs | 47,846 | 2,373 | 1,617 | 96 | 97,746 |
| Exons | 277,161 | 325 | 142 | 2 | 38,823 |
| coding | 260,368 | 299 | 140 | 2 | 38,823 |
| non-coding | 27,774 | 515 | 152 | 9 | 36,521 |
| Introns | 247,597 | 1,355 | 160 | 30 | 611,280 |
| coding | 235,861 | 1,207 | 152 | 30 | 611,280 |
| non-coding | 22,579 | 2,911 | 304 | 30 | 498,241 |

mysteries and aids in answering other unknown features, such as the effects of sexual selection and mate choice systems on genome evolution.

## DATA AVAILABILITY

The genome is available on NCBI with the assembly accession number GCA_024217435.2. The genome is annotated via the NCBI eukaryotic genome annotation pipeline, and the annotation report release (100) is available here. Several smaller contigs and contaminant microbes were removed in the annotation pipeline yielding a more robust genome assembly. The sequence identifier for the chromosome-level scaffolds is available in the GigaDB [28]. The NCBI Bioproject accession number is PRJNA851781, the raw Hi-Fi sequence accession is SRR19820733, the Hi-C sequence accession is SRR22219025, and the RNA-Seq sequence files from various tissues are SRR20438584–SRR20438604. Additional data is available in the GigaDB [28].

## DECLARATIONS

## List of abbreviations

BUSCO: Benchmarking Universal Single-Copy Orthologs; CCS: Circular Consensus Sequence; Gb: Giga basepair; Hi-Fi: High-Fidelity; Mb: Mega basepair; NCBI: National Center for Biotechnology Information; not: nucleotide; QUAST: Quality Assessment Tool; QV: Quality Value; SMRT: Single Molecule Real Time; University of Oregon Genomics and Cell Characterization Core Facility (UOGC3F).

## Ethical approval

Not applicable.

## Consent for publication

Not applicable.

## Competing Interests

The authors declare that they have no competing interests.

## Funding

This work was funded by National Science Foundation (NSF) Grant 2015419 to WAC and AGJ and Grant 1953170 to AGJ. We also acknowledge the startup funds provided by the University of Idaho to AGJ.

## Authors' contributions

Author contributions, described using the CRedIT taxonomy are as follows:

Conceptualization: BR, CMS, SB, WAC, AGJ; Methodology: BR, CMS, SB, BDJ, EB; Software: BR, CMS, HH, MC; Validation: BR, CMS; Formal Analysis: BR, CMS; Investigation: BR, CMS; Resources: MC, BDJ, EB, MM; Data Curation: BR, CMS, MC; Writing – Original Draft Preparation: BR, CMS, AGJ; Writing – Review & Editing: BR, CMS, AGJ; Visualization: BR, CMS; Supervision: WAC, AGJ; Project Administration: CMS, SB, WAC, AGJ; Funding Acquisition: WAC, AGJ.

## Acknowledgements

We are truly grateful for the dedicated efforts of Emily Rose and her students at Valdosta State University, who collected the *S. scovelli* samples crucial for this work (Florida Fish and Wildlife Conservation Commission Permit: SAL-17-0182-E, SAL-18-0182-E). We also thank Jeff Bishop and Tina Arredondo from the University of Oregon (UO) GC3F for library preparation and sequencing assistance. We want to acknowledge the staff at Phase Genomics for their helpful Hi-C technical support. We are grateful to Mike Coleman and Mark Allen for assisting with Talapas Supercomputer Cluster at UO and Benji Oswald with Research Computing and Data Services at UI. We greatly appreciate the support of the NCBI staff for the Eukaryotic Annotation Pipeline. We thank Jacelyn Shu for her pipefish illustrations. We are grateful for the helpful comments and review by Sven Winter and Yue Song on the manuscript.

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
