## [Reviewer Report]

Comments on revised manuscriptThank you for the improvement of the manuscript. It is now easier to follow and includes more information as before. It was a bit difficult to see the changes as they were not highlighted and the lines are not numbered.  Despite that, I have only a few minor comments that should be addressed easily so that the manuscript will be ready for publication soon.   Line numbers in the comments refer to lines of the specific paragraph/section.  DNA and RNA extraction: L7:such as? If you listed all tissues, please remove such as, if you sequenced RNA for nor tissues please add them.   Sequencing and Assembly: L5: 159 bp is an uncommon read length. Was this just a typo, or how did that come to be? L10: remove "the" before juicer; otherwise, it sounds like an actual fruit juicer instead of a bioinformatics tool ;-). Same for 3D-DNA in the line below. Please make it more clear in the text if you sequenced the RNA for each tissue separately or in one library.   L11-12: I am not convinced that not allowing for correction was the right approach. Did you test how the results would look with corrections enabled?  Assembly Statistics and Quast Results:  Quast calculates assembly statistics so I am not sure why the header needs to include both.  L5: Please avoid using "better" but instead rephrase so that is is clear that the NG50 is 1.75x larger than the previous assembly. "Better" is not clear.  Busco and Merqury results:  I would not claim that Busco says the genome is 95% complete, as busco only tries to find genes that are supposedly orthologous in Actinopterygii. So I would rather say Busco suggests a high completeness as it finds 95% of the orthologs. Also, all genes in the Busco dataset are supposed to be single-copy orthologs; therefore, I would not say that 93% are conserved single-copy orthologs, as the remaining duplicated or fragmented genes could just be assembly errors.   Please also state the Merqury QV value, and I would suggest stating the error rate in %.   I still find the discussion about missing Busco genes strange, as since Busco 4 or 5 the datasets all got much larger and the Busco completeness values went down in most assemblies, even in well studies taxa as mammals. With recent datasets, it is very unlikely to get much more than 95-97%. In my opinion, it is rather a sign of too large and incorrect Busco datasets than evidence for missing orthologs. I would at least add that point to the discussion.  Table 1: Please follow standard practice in scientific writing and add separators to the numbers in all tables (main text and supplementary), e.g., 28444102 → 28,444,102. Otherwise, they are difficult to read.   Annotation Results:  L3: 20,101 coding genes, 18,616 genes … Please check throughout the whole manuscript for consistent style.   Data Availability: L2: Annotation report release 100. What does "100" stand for? Also, "at here" sounds not correct; please remove "at".  L4: Table S2 does not show the scaffold identifiers. L5: please state the complete BioProject accession not just the numerical part.    Supplementary data:   Please change numbers in all tables to standard format e.g., 21,671,036

---

## [Reviewer Report]

Reviewer name and names of any other individual's who aided in reviewer Sven WinterDo you understand and agree to our policy of having open and named reviews, and having your review included with the published papers. (If no, please inform the editor that you cannot review this manuscript.)YesIs the language of sufficient quality?YesPlease add additional comments on language quality to clarify if needed
The writing style is quite different to standard scientific English. I would suggest a nearly complete rewrite to make it more concise and more structured.Are all data available and do they match the descriptions in the paper? YesAdditional CommentsAre the data and metadata consistent with relevant minimum information or reporting standards? See GigaDB checklists for examples <a href="http://gigadb.org/site/guide" target="_blank">http://gigadb.org/site/guide</a>NoAdditional CommentsI am missing detailed sampling locations and permit information.Is the data acquisition clear, complete and methodologically sound?NoAdditional CommentsPermits and sample locations are missing. Is there sufficient detail in the methods and data-processing steps to allow reproduction?NoAdditional CommentsI am missing more detailed QC steps and, in general, more details about the methods that were used.  How much Hifi data was generated in Gb?  Which k-mer size was used for genome size estimation? Was Trimmomatic only used to trim the first base? What is a first-pass genome assembly?  What does "with no breaking of original contigs" mean? How can you correct and manually curate an assembly without breaking contigs? Was this a statement or a setting? Why was there no polishing and gap-closing performed? I understand the assembly is based on hifi-reads I would at least mention that no polishing is needed if that is the case, or did hifiasm include a polishing step?  There is no explanation of the annotation process or the RNA sequencing.  When comparing N50 values it is important to use NG50 instead.
Is there sufficient data validation and statistical analyses of data quality? NoAdditional CommentsI am missing more QC steps, such as Blobtools, Merqury, etc. to properly validate the accuracy of the assemblyIs the validation suitable for this type of data?YesAdditional CommentsThe steps that were done, yes, but they are insufficient, in my opinion. Is there sufficient information for others to reuse this dataset or integrate it with other data?YesAdditional CommentsI would rather say that there are to many unnecessary tables with no real information in it.  List of missing Busco genes? There is no need for that as they are likely just missing due to the assembly quality.   List of Accession numbers of Chromosome-level scaffolds without information about what chromosome the accession number belongs to. It is also very strange that the chromosomes are not ordered by length and that they are not listed as chromosomes on NCBI. Any Additional Overall Comments to the AuthorI am really sorry, and I do not want to sound mean, but this manuscript needs major improvements in structure, writing, and data validation. It violates so many standard practices of scientific writing.   I have never seen anybody cite a full title of a previous manuscript. There is absolutely no need for that. The annotation is labeled as an improved annotation, but its results are only listed in the abstract, and it is not mentioned how it is generated anywhere other than the data availability section. That the genome is tagged under RefSeq by NCBI is absolutely unnecessary information in the abstract, this is just a label, and it tells not much about the quality.  I would urge the authors to restructure the manuscript. Start with a short description of the species and why the species and its genome is important as an introduction, then focus on a detailed data description with methods and basic results such as assembly statistics (importantly not just scaffold N50 but also on the contig-level!), Busco, Merqury completeness and error rate, genome size estimate, annotation (repeat and gene), etc.  There is really no need for 30 pages of useless supplementary tables (please also make sure that next time you sort the files during the submission so that the pdf does not start with 30 pages of tables). The data cannot support any information about gene loss, as there is so much of the assemblies not properly anchored into chromosomes.  I would also try to improve the Hi-C contact map figure. There is really no need for the blue and green boxes and the assembly label at the x-axis. I may have overlooked it due to the writing style, but I would like to see mentioned how much of the assembly is in the chromosome-scale scaffolds and how much is unplaced.   I like the improved assembly, it just needs a much better presentation in form of a well-structured manuscript, and unfortunately, in its current form, it clearly is not well-structured. There are plenty of other data notes available as templates. I personally would always opt for a more traditional manuscript structure (Introduction, Methods, combined Results and Discussion), but that is my personal preference. I hope my comments are helpful, and I am looking forward to seeing a revised version in the future. RecommendationMajor Revision

---

## [Reviewer Report]

Reviewer name and names of any other individual's who aided in reviewer Yue SongDo you understand and agree to our policy of having open and named reviews, and having your review included with the published papers. (If no, please inform the editor that you cannot review this manuscript.)YesIs the language of sufficient quality?YesPlease add additional comments on language quality to clarify if needed
Are all data available and do they match the descriptions in the paper? YesAdditional CommentsAre the data and metadata consistent with relevant minimum information or reporting standards? See GigaDB checklists for examples <a href="http://gigadb.org/site/guide" target="_blank">http://gigadb.org/site/guide</a>YesAdditional CommentsIs the data acquisition clear, complete and methodologically sound?YesAdditional CommentsIs there sufficient detail in the methods and data-processing steps to allow reproduction?NoAdditional Commentsneed more detailed paramaters and process about genome assembly. Although using the NCBI pipeline for gene annotation, it is better to give more details too.Is there sufficient data validation and statistical analyses of data quality? YesAdditional CommentsIs the validation suitable for this type of data?YesAdditional CommentsIs there sufficient information for others to reuse this dataset or integrate it with other data?YesAdditional CommentsAny Additional Overall Comments to the Author(1) Please state clearly how much CCS Hi-Fi data has been produced by sequencing and hic-data finally used for chromosome assembly after filtration, not just the number of reads.  (2) Please state clearly the estimated genome size using Hi-Fi data.  (3) What is the process for “correct primary assembly misassembles”? Please described in detail. (4) In Table 1, I noticed that the difference between the new and previous genome of S.scovelli is more than 100M (about 25% of the size of the newly assembly). Otherwise, most of genome size of Syngnathus species ranged from 280-340 Mb, I think take some explanation of these extra sequences is necessary. (5) Need more detailed paramaters and process about genome assembly and gene annotation. (6) Whether the previous version had any assembly errors and updated in this new one. if exists, please state clearly. RecommendationMinor Revision